# Effectiveness and Factors Associated with Improved Life Skill Levels of Participants of a Large-Scale Youth-Focused Life Skills Training and Counselling Services Program (LSTCP): Evidence from India

**DOI:** 10.3390/bs12060191

**Published:** 2022-06-15

**Authors:** Gautham Melur Sukumar, Swati S. Shahane, Anusha B. Shenoy, Srividya Rudrapattana Nagaraja, Prathyusha P. Vasuki, Prathap Lingaiah, Shalini Rajneesh, Pradeep S. Banandur

**Affiliations:** 1Department of Epidemiology, Center for Public Health, National Institute of Mental Health and Neuro Sciences (NIMHANS), Bengaluru 560029, India; drgauthamnimhans@gmail.com; 2Life Skills Project, Department of Epidemiology, Center for Public Health, National Institute of Mental Health and Neuro Sciences (NIMHANS), Bengaluru 560029, India; swati.shahane@gmail.com (S.S.S.); anushabshenoy@gmail.com (A.B.S.); rnsri93@gmail.com (S.R.N.); 3Department of Biostatistics, National Institute of Mental Health and Neuro Sciences (NIMHANS), Bengaluru 560029, India; vasukiusha1987@gmail.com; 4State NSS Wing, Government of Karnataka, Bengaluru 560001, India; gokstateliaisonofficer@yahoo.com; 5Planning Department, Government of Karnataka, Bengaluru 560001, India; shalinirajneesh.sr@gmail.com

**Keywords:** life skills, mental health, Yuva Spandana, youth, training

## Abstract

**(1) Background:** To empower and facilitate mental health promotion for nearly 18 million youth, a pioneering state-wide Life Skills Training and Counselling Services Program (LSTCP) was implemented in Karnataka, India. This study assesses the changes in life skills scores, level of life skills and factors associated with increased life skills among participants of the LSTCP. **(2) Method:** This pre–post study design was conducted on 2669 participants who underwent a six-day structured LSTCP. Changes in mean life skills scores and level of life skill categories pre- and post-LSTCP were assessed. Multivariate logistic regression was performed to assess the factors associated with increases in life skills. **(3) Results:** The LSTCP resulted in significant changes in life skill scores and level of life skills, indicating the effectiveness of the training. All life skill domains, except empathy and self-awareness, increased post-training. There was a positive shift in the level of life skills. Age (AOR = 1.34, CI = 1.11–1.62), gender (AOR = 1.39, CI = 1.15–1.68), education (AOR = 1.44, CI = 1.05–1.97) and physical (AOR = 1.02, CI = 1.01–1.03) and psychological (AOR = 1.02, CI = 1.01–1.03) quality of life was associated with an increase in life skills among participants. **(4) Conclusions:** The LSTCP is effective in improving the life skills of participants. The LSTCP modules and processes can be used to further train youth and contribute to mental health promotion in the state.

## 1. Introduction

Youth account for 27.5% of the population in India [1], and 34.6% in the state of Karnataka [2]. Ensuring the best possible physical and mental health of youth is a key national investment and a constitutional mandate [3]. Fuelled by macro-level determinants such as globalisation, urbanisation, industrialisation, technology revolution and dwindling traditional social support systems, the 21st century youth in India are vulnerable to a unique set of health and social issues. A literature review indicated that undernutrition (56.4–68.5%), micronutrient deficiency (25%), overweight (9.9–19.9%), common mental disorders (20%), stress (20%), suicides (3.73–3.96 per 100,000 population per year), tobacco use (45%), harmful alcohol use (21.4%), hypertension (10.1%), road traffic injuries (29%) and violence (27%) are some of the prominent health issues among youth in India, with significant inter-state variations [4].

To address youth-specific health issues, a youth-driven, state-wide mental health promotion programme named ‘Yuva Spandana’ was implemented in the State of Karnataka, India, on 26 March 2014 [5]. This programme established wellness centres in 30 districts of the state and reached around 7,214,223 youth. Further programme-related detailsare available at www.yuvaspandana.in (accessed on 26 May 2022) [6]. Many of the issues mentioned above are behavioural. Yuva Spandana, along with life skills training, help to mould the behavioural characteristics of trained youth as well as beneficiaries.

The mid-term external evaluation of the Yuva Spandana programme [7] and ensuing discussions to strengthen implementation of youth policy directives by the Government of Karnataka revealed the need for implementing an effective large-scale life skills training programme for youth through their teachers/university faculty/influencers as a long-term intervention for the prevention and control of youth health issues. The recommendation was to develop an experiential learning model to equip youth with 10 essential life skills (self-awareness, empathy, effective communication, interpersonal relationships, coping with stress, coping with emotions, decision making, problem solving, creative thinking and critical thinking), as recommended by UNICEF, New York, NY, USA [8] and the WHO, Geneva, Switzerland [9].

To implement the above recommendations, the National Service Scheme (a program for youth within universities in India) of the Department of Youth Empowerment and Sports, Government of Karnataka, partnered with the Department of Epidemiology, Centre for Public Health (CPH), National Institute of Mental Health and Neuro Sciences (NIMHANS), Bengaluru, and developed the Life Skills Training and Counselling Services Program (LSTCP) for youth in Karnataka.

Phase-1 of this program aims to empower and bring positive change to the lives of youths through trained NSS officers/coordinators, faculty from pre-university, collegiate education, technical education and universities in the state of Karnataka. In phase-2 of the program, the trained individuals would further train youth in their respective instititutes and also facilitate access to psychological services to youth in need.

Evidence worldwide indicates the proven effectiveness of life skills training to bring positive change amongst vulnerable populations (substance users [10], juvenile delinquents [11], migrants [12] and sex workers [13]), but its effectiveness within an apparently healthy adult population has not been clearly established, especially in the Indian context.

The LSTCP is the first such large-scale life skills and positive mental health development training programme on apparently healthy individuals in India. It is a unique inter-sectoral initiative because the departments of youth empowerment, education, higher education, and health have joined hands to provide youth mental health promotion services. As of 24 February 2020, nearly 2669 faculty have been trained in the LSTCP. 

In the above context, this paper aims to assess the change in life skill scores and level of life skills among participants of the LSTCP. It also aims to identify factors associated with increased life skills among participants of the LSTCP.

## 2. Materials and Methods

This pre–post study design was conducted on a sample of 2669 participants (National Service Scheme (NSS) coordinators, officers and teaching and non-teaching faculty of different universities across Karnataka) who underwent a six-day structured residential life skills training (intervention) between January 2017 and February 2020. The participants of the LSTCP program were deputed mostly from within the government setup, namely directorates of collegiate education, technical education, pre-university board and 48 universities across 30 districts of Karnataka. Deputation of participants by their respective authorities took place on request by interested participants on a first-come-first-served basis.

The six-day experiential learning and facilitatory method-based life skills training was modelled on the principles of Kolb’s learning theory [14] and includes ten World Health Organization (WHO) recommended essential life skill domains [15]. Details of the training are available in the Life Skills Training Manual [16]. A manuscript on the methodology of the LSTCP has recently been submitted for publication.

During the training program, a pre- and post-training assessment of level of life skills was conducted using a semi-structured self-administered questionnaire. The pre-test questionnaire had 25 sections, while the post-test questionnaire had 12 sections (see Table 1). Sections 3, 5, 6, 7, 12, 13, 14, 15, 16, 17, 18, 19 and 21 collected information regarding participant’s socio-demographic, morbidity and behaviour-related information, which are unlikely to change immediately post-training. Hence, these sections were not included in the post-test questionnaire. Items specific to life skills assessment were included in both the pre- and post-test questionnaires. Details of sections, number of items and the scoring adopted in the questionnaire is provided in Table 1.

After providing informed consent, participants filled out the self-administered, bilingual questionnaire immediately before and after the training programme, which was conducted at NIMHANS, Bengaluru. The interview method was not adopted as participants were educated and comfortable with self-administered questionnaires, but project staff assisted in clarifying questions or doubts during the filling out of the questionnaires. Project staff invigilated the entire process and repeatedly instructed participants not to discuss the content of the questionnaires with each other while filling them out, to reduce information bias.

Post-hoc power testing to detect differences in life skills scores and levels and factors associated with life skill improvement was carried out using Open-Epi and was found to be more than 95% for our study sample of 2669 participants. All data collected were kept confidential and stored in a locked safe. Data were entered on a specifically designed password-protected database. Only selected study staff had access to these data.

The cleaned dataset was analysed using STATA 12.0. Numerical and categorical data were described using mean (SD) and percentages, respectively. Normality was tested using the Shapiro–Wilk test (*p* > 0.05 considered normal distribution). Significant changes in mean life skills scores pre- and post-training were tested using the paired-*t* test. Changes in the level of life skills categories (moderate and high level) pre- and post-training were assessed through the McNemar–Bowker test (*p* < 0.05 was considered significant).

The life skill scores pre- and post-training were computed, and participants who had an increase in life skills scores post-test were categorised as “Participant with increased life skills scores”. Any participant with increased life skills score (Yes and No) was considered as a binary dependent outcome variable. Multivariate binary logistic regression was performed to assess factors associated with increase in life skill scores.

The independent variables in the model (age, gender, marital status, education qualification, substance use, personality factors and quality of life) were selected according to a conceptual framework developed based on a literature review and expert opinion (Refer to Appendix A). All independent variables significant at the 10% level in univariate analysis were included in the final multivariate model. The final multivariate model was developed using a forward stepping process. All variables significant at the 5% level and those that changed the odds ratio of at least one preceding variable by at least 10% were eligible for retention in the final multivariate model. The model fit was assessed using the command *estat gof in STATA 12.0* for goodness of fit. The model had to be fit (Hosmer–Lemeshow value = 0.62) to be considered for analysis. Area under the curve was assessed using the *lroc* command. All the analysis was carried out using Stata 12.0 software for windows.

Ethical approval was obtained from the Institutional Ethics Committee at NIHMANS, Bengaluru vide letter No. NIMHANS/2 ND IEC (BS & NS DIV.)/2016 dated 7 December 2016.

## 3. Results

Of the 2669 participants, nearly 69% were male, 58% were urban residents, 99% were Hindu by religion, 91% had a post-graduate education and 78% were currently married. The mean age of the participants was 39.7 years.

A significant difference in mean life skill scores was observed between pre- and post-training (mean difference = 7.71 points, *p* < 0.05). There was a significant increase in the proportion of participants with increased life skills scores post-training (58%, *p* < 0.001). The proportion of participants with increased life skills was strongly associated with gender (*p* = 0.003), religion (*p* = 0.02), education (*p* = 0.004) and marital status (*p* = 0.04). The maximum difference in mean scores post-training was observed among participants in non-salaried employment (25.61), not living with spouse (13.29) and among youth (12.59) (see Table 2).

There is a significant mean difference between pre- and post-training life skill scores (*t* = 10.843, *p*-value = <0.001). Likewise, there is a significant increase in the mean scores post-training in the domains of decision making (*t* = 2.5009, *p*-value = 0.0124), problem solving (*t* = 9.569, *p*-value = <0.001), communication (*t* = 13.119, *p*-value = <0.001), interpersonal relationship (*t* = 6.932, *p*-value = <0.001), coping with emotions (*t* = 5.351, *p*-value = <0.001), coping with stress (*t* = 12.655, *p*-value = <0.001), creative thinking (*t* = 17.754, *p*-value = <0.001) and critical thinking (*t* = 3.667, *p*-value = 0.0003). However, there is a significant decrease in the mean score of empathy (*t* = −30.1147, *p*-value < 0.001) and self-awareness (*t* = −9.088, *p*-values < 0.001) after life skills intervention (Table 3).

Nearly 39% of people with a moderate level of life skills at pre-training converted to a high level of life skills post-training. Similarly, nearly 88% of participants with a high level of life skills pre-training continued to remain high post-training. The change in post-training is significant (*χ*^2^ = 19.41, *p* = <0.001), and there was a positive shift in the level of life skills of the participants (Table 4).

Table 5 shows that participants aged between 18 and 35 years (OR = 1.41, CI = 1.18–1.70); females (OR = 1.32, CI = 1.10–1.58); participants’ who were married (OR = 0.79, CI = 0.64–0.98); participants who had completed a degree/diploma (OR = 1.56, CI = 1.15–2.12); participants who had personality traits which included extroversion (OR = 1.13, CI = 1.03–1.24), agreeableness (OR = 1.1, CI = 1.00–1.19), conscientiousness (OR = 1.09, CI = 1.00–1.18) or neuroticism (OR = 0.87, CI = 0.79–0.96) and all domains of quality of life were significantly associated with an increase in the life skills of participants.

Multiple logistic regression analysis showed that participants aged between 18 and 35 years (AOR = 1.34, CI = 1.11–1.62), females (AOR = 1.39, CI = 1.15–1.68), participants with postgraduate education (AOR = 1.44, CI = 1.05–1.97) and participants with better physical quality of life (AOR = 1.02, CI = 1.01–1.03) and psychological quality of life (AOR = 1.02, CI = 1.01–1.03) had higher odds for an increase in life skills (see Table 6).

## 4. Discussion

The Life Skills Training and Counselling Services Program resulted in significant changes in life skill scores and level of life skills before and after the training, indicating the effectiveness of the training. All life skill domains except empathy and self-awareness increased post-training, and there was a positive shift in the level of life skills. Age (18–35 years), gender (female), education (post-graduation and above), physical quality of life and psychological quality of life are associated with an increase in life skills among participants attending the life skills training program.

The LSTCP is a first-of-its-kind program in India, for youth mental health promotion implemented state-wide through a large team of qualified life skill trainers, delivered as an intersectoral model. The knowledge gained from this programme has far-reaching implications on youth health programme implementation across the country, and especially conveys the need for a paradigm shift towards an inter-sectoral department-driven programme rather than uni-department-driven youth health programs. Lessons could also be applicable to similar countries.

The LSTCP has bridged the evidence gap regarding the effectiveness of training programmes and factors associated with increased life skills, which are limited in the Indian setting. Though epidemiologically not comparable, several studies on vulnerable populations and in sports and school contexts also showed a similar increase in life skills [25,26,27,28,29,30]. The comparability of our study findings with other similar trainings on apparently healthy participants in India is limited as there is no such documented evidence available.

Decision making, problem solving, communication skills, interpersonal relationship skills, coping with emotions, coping with stress, creative thinking and critical thinking were increased post-training, as proven previously by different studies on school students [31]. The fact that the LSTCP training was conducted in groups of 25 participants and was delivered through an experiential learning process resulted in better involvement and interaction between facilitators and participants. This could have contributed to improved life skills. In contrast, self-awareness and empathy scores decreased post-training; regarding this, no studies have been found which support similar results on an apparently healthy population. Barlow et al. [32] states that life skills training leads to actual knowledge and an individual’s perception of their capabilities. Hence, the training grounds the participants and makes them accept themselves as they are rather than having an ideal image of themselves, which can help them in dealing with inevitable problems with comprehensive knowledge. This might explain the decreased post-training scores for self-awareness and empathy among participants.

Life skills scores were increased by an average of 7 points across all subcategories after training. The highest increase in pre–post difference was observed in youth, females, participants with a post-graduation degree or diploma and those who were not living with a spouse. Life skills increased more in youth, as also observed in other studies [33]. The reasons for this could be attributed to the curious nature of youth [34] and their eagerness and interest to learn new things [35], along with the fact that they are more participative in experiential learning processes [36]. It also indicates that youth understood and filled out the questionnaires appropriately, further strengthening the validity of the instrument.

Though post-training average scores were the same across all age groups, youth demonstrated a higher increase because their baseline scores were comparatively lower, indicating that this programme is more effective in improving life skills for this age group. Being a youth-focused life skills training program, which is subsequently scheduled to be implemented to youth by trained participants (NSS officers, coordinators, teaching faculty), the results indicating increased effectiveness in youth is very encouraging. The programme converted 39% of participants from moderate to high life skills and retained 88% of participants at high life skills after training.

Unlike other studies [37], our study observed that females benefitted more than males in adapting life skills. Very few studies [38] have supported similar results. Generally, women tend to be more extravert [39], where they communicate more, express more, experience positive emotions more and empathize more with others compared to males. Participants who have completed their post-graduation and above are likely to experience high life skills through the training program. According to Earle [40], the higher the level of literacy, the greater opportunities there are to earn high income, which will reflect on the skills and knowledge applied to the job. In this training, most of the graduates were lecturers in a government organization with a secure job, and their interest in learning more about life skills might have contributed to them developing high life skills. Participants with better physical and psychological quality of life scores benefitted more in adapting life skills, as suggested by several studies [37,41,42,43], in contrast to some studies where life skills training had higher variables in all domains of quality of life, except physical health [44]. Training helps individuals to understand their strengths and weaknesses in order to have a better physical quality of life. The training sessions were carried out in groups, which can have a positive impact on reducing stress, thus minimizing the negative mood resulting in accepting and dealing with reality [45]. This contributes to increased psychological quality of life.

The LSTCP training was delivered by highly trained and experienced resource individuals, which is also a factor that could have influenced the improvement in the life skill scores of participants. In the current stage of the project, there is limited evidence regarding the extent of improvement in life skills among the youth who have been trained by these trained participants (faculty) at NIMHANS. However, the project investigators believe downstream trainings would also be equally effective as they are based on experiential learning methods, participative in nature, and there is a detailed manual for reinforcement of trainings to ensure sufficient reliability in the quality of further training.

Data from secondary sources indicate that around 183,666 teachers in high school and pre-university colleges [46] and approximately 5200 faculty at degree colleges are present in Karnataka state, which has a youth population of nearly 18,600,000 [2]. With 2669 staff trained in LSTCP, we have 0.14 trained staff for every 1000 youth in the state. With mental health and behavioural issues on the rise among youth, there is a high need for psychosocial first aid, which can be provided by these trained officers. The LSTCP provides the necessary access, hereby complementing the efforts of the health department. It is therefore a unique inter-sectoral attempt to empower as well as deliver youth mental health promotion services in the state.

### Limitation

The study is not without limitations. Although the selection of participants was based on deputation, there is a considerable geographic representation of participants from across the state. We expect that these deputed officers are no different from those who were not deputed. Hence, we feel that the influence of selection bias related to outcome is either unlikely or negligible. However, to our knowledge, supporting evidence for the same is not available in the current existing literature. Data collection using self-administered questionnaires offers limited control over the responses provided, as well as the order in which the respondent fills the questionnaire. However, the presence of one of our project team members to facilitate respondents while filling out the questionnaire as well as providing clear instruction and informed consent prior to questionnaire administration is likely to minimize this limitation. The presence of a team member was also to clarify the doubts of the participants if they had any, and there was no pressure/forcing of respondents for desirable answers in favour of the study. The highest level of control over the questionnaire was with the participants as it was a self-administered questionnaire, hence interviewer bias was reduced.

## 5. Conclusions

The youth-focused Life Skills Training and Counselling Services Program (LSTCP) was effective in improving the capacities and life skills of the participants. Participants who were youth (18–35 years), female and post-graduate degree holders, and those with a better physical and psychological quality of life, are more suitable for this programme. The LSTCP is at the stage of upscaling this training to nearly 5000 potential faculty and provides life skills training and mental health promotion services for nearly 20 million youth across Karnataka state. There is also scope for expansion in other languages and to other states in India and the potential to contribute towards promoting mental health among youth across the country.

### 5.1. Implication

The Life Skills Training and Counselling Services Program provides evidence to the fact that youth are more receptive to improvement. The deputation of more young adults to the program may strengthen their life skills for their future. Considering our study results, females need to be deputed more. Expanding the program across other sectors, including industries/IT, would help to increase life skills among youth. As mental health care is plagued by stigma and limited access in rural areas, this program will provide a platform for youth from rural backgrounds to play a facilitatory role in providing mental health care in India. Considering the number of youth in the state, there is a high need to increase the number of trained teachers for youth mental health promotion. Additionally, introducing life skills training and counselling services programs as early as possible in life may reduce the risk of developing mental health problems.

### 5.2. Implementation for the Future

The Life skills Training and Counselling Services Program, at its current stage, is focusing on developing the capacity of faculty to identify and train youth in their respective institutions to provide mental health promotion services. Subsequently, the trained youths develop linkages to work as youth volunteers under the Yuva Spandana program, a national health mission to increase the accessibility and coverage of mental health promotion services in the state. There is scope for research to understand the effectiveness of such interventions and their tangible and intangible effects on the prevalence of the behavioural disorders of youth.

## Figures and Tables

**Table 1 behavsci-12-00191-t001:** Details of pre- and post-test questionnaire for assessment of life skills among participants attending the LSTCP.

Section Number	Section Name	Assessment Focus and Scale Used	Number of Items	Operational Definition and Scoring Category
1	Interview Information ^#,^*	Auto-assigned unique ID, Name, Address, Date and place, Consent information	17	NS
2	Editing and Data entry ^#,^*	Details of data entry	12	NS
3	Socio-demographic characteristics ^#^	Socio-demographic characteristics of the respondents, including number of family members, age, occupation, education and marital status	13	NS
4	Family environment ^#,^*	Communication with family members, Arguments, Criticism, Time spent with family members, Decision making and Family support issues	20	NS
5	Socio-economic characteristics ^#^	Ownership of house, agricultural land and livestock. Monthly income and expenditure.	7	NS
6	Personal and family health ^#^	Morbidity and hospitalisation information of self and family members	10	NS
7	Diet and eating habits ^#^	Type of diet and dietary practises	3	NS
8, 9, 10, 11	Substance Use *	Tobacco smoking	9	**C.A.G.E. Questionnaire [17]—Current smokers**: defined as persons who reported smoking at least 100 cigarettes during their lifetime and those who reported smoking every day or some days.
				A score of two ‘yes’ responses constitutes a positive screening test
		Tobacco chewing	9	Fagerstrom scale [18]—Chewing tobacco at least once during their lifetime and those who reported chewing tobacco every day or some days, at time of interview. Among users, dependents were persons with the scores below
				Score
				<4—moderate dependence
				5+—significant dependence
		Alcohol use and dependence	16	M.I.N.I—5.0.0 scale [19]. A problematic pattern of alcohol use leading to clinically significant impairment or distress, as manifested by at least two of the DSM criteria occurring within a 12-month period. A score of three or more ‘yes’ responses on the first 7 questions indicates current alcohol dependence. A score of one or more ‘yes’ responses to the last 4 questions indicates current alcohol abuse
		Injecting/sniffing/oral drugs	7 (Screening questions and scale items)	C.A.G.E—AID [20] is a 4-item scale where a score of two ‘yes’ responses indicates a possible substance use disorder and a need for further testing.
12	Violence-related information ^#^	Experience of violence, Types of violence experienced, Hospitalisation due to injuries, Violence inflicted: person involved	6	NS
13	Depression	Symptoms of depression	9	M.I.N.I—5.0.0 Scale [19]—Five (or more) of the DSM symptoms should have been present during the same 2-week period and represent a change from previous functioning; at least one of the symptoms is either (1) depressed mood or (2) loss of interest or pleasure. A score of 5 or more ‘yes’ responses indicates major depressive episodes
14	Generalised Anxiety Disorder	Symptoms of anxiety	7	M.I.N.I—5.0.0 Scale [19]—Excessive anxiety and worry (apprehensive expectation), occurring for more days than not for at least 6 months, concerning a number of events or activities (such as work or school performance). A score of 3 or more ‘yes’ responses indicates generalised anxiety disorder
15	Self-harm	Suicidal ideation and behaviour	16	M.I.N.I.—5.0.0 Scale [19]—This scale assesses risk of suicide. Further action needs to be taken.One or more ‘yes’ responses indicates suicide risk
				1–8 points—Low
				9–16 points—Moderate
				>17 points—High
16	Injuries and related ^#^	Injury experiences, Types of injuries experienced	3	NS
17	Physical Activity ^#^	Time spent on different physical activities	3	NS
18	Sexual Behaviour ^#^	Duration of Sexual activity, Number of partners, Use of condoms	6	NS
19	Work Environment and Job Satisfaction ^#^	Type of organisation, Change of job, Job satisfaction, Duration of outdoor stay at work	23	NS
20	Teaching Factors ^#,^*	Mode of teaching, Preferred mode of teaching, Perception about teaching abilities, knowledge, technology	8	NS
21	Peer group and social capital ^#^	Number of peers, Activities performed with peers, Peer characteristics	8	NS
22	Behavioural Factors ^#,^*	Self-talk, Crisis related information, Personality factors	46	Personality factors (Big Five Inventory-10) [21]—Two questions are scored together for 5 different personality factors
23	Life skills *	Life skills of an individual	115	Life Skills scale [22]
				≤397—Low Life skills
				398–437—Moderate Life skills
				438+—High Life skills
24	Quality of Life *	General health and physical, psychological, social and environmental quality of life	26	WHO-Quality of Life—BREF [23]. Score range 4–20. Converted to percentile score. The higher the score, the higher the quality of life
25	Exposure to Media and related ^#,^*	Information related to usage, time spent, kinds of programs, reasons to watch TV, Internet, Video tape, Video games, Mobile phone usage and risk of addiction	12	Cell phone overuse and addiction [24]. A score of 3 or more ‘yes’ responses indicates a risk of cell phone addiction

* These 12 questionnaires were also included in the post-test questionnaire. ^#^ Questionnaires were developed for the present study. NS—Not Scored.

**Table 2 behavsci-12-00191-t002:** Life skill scores (pre–post training) and participants with increased life skills score post-training by select socio-demographic variables.

X	Life Skills Scores			Increased in Life Skills Score Post-Training			χ2	*p* Value
	Pre-Training	Post-Training	Mean Difference	Yes	No	Total		
	Mean ± SD	Mean ± SD		*n* (%)	*n* (%)	*N*		
Number of participants	455.67 ± 40.84	463.37 ± 41.25	7.71 ^#^	1351 (58.16)	972 (41.84)	2323 (100.00)	10.84 ^$^	<0.001 *
**Age in years (*n* = 2306)**								
18–35	450.69 ± 41.98	463.28 ± 44.22	12.59 ^#^	485 (63.90)	274 (36.10)	759	15.3	<0.001 *
36–50	458.09 ± 39.06	463.49 ± 39.31	5.41 ^#^	728 (55.61)	581 (44.39)	1309		
51 and above	458.71 ± 43.12	463.14 ± 41.89	4.44 ^#^	129 (54.20)	109 (45.80)	238		
**Locale (*n* = 2323)**								
Rural	463.17 ± 41.44	454.58 ± 40.64	8.59 ^#^	577 (60.23)	381 (39.77)	958	2.88	0.09
Urban	456.43 ± 40.97	463.52 ± 41.12	7.08 ^#^	774 (56.70)	591 (43.30)	1365		
**Gender (*n* = 2323)**								
Male	455.75 ± 41.56	462.14 ± 41.83	6.38 ^#^	902 (56.09)	706 (43.91)	1608	9.14	0.003 *
Female	455.46 ± 39.21	466.15 ± 39.79	10.69 ^#^	449 (62.80)	266 (37.20)	715		
**Religion (*n* = 2323)**								
Hindu	455.86 ± 40.47	464.07 ± 40.43	8.2 ^#^	1266 (58.91)	883 (41.09)	2149	5.84	0.02 *
Others	453.04 ± 45.56	455.11 ± 49.90	2.06 ^#^	84 (49.41)	86 (50.59)	170		
**Education (*n* = 2323)**								
Post-graduation and above	456.45 ± 40.99	463.76 ± 41.44	7.3 ^#^	1211 (57.23)	905 (42.77)	2116	8.38	0.004 *
Till Degree/Diploma	447.64 ± 38.43	459.46 ± 39.13	11.82 ^#^	140 (67.63)	67 (32.37)	207		
**Occupation (*n* = 2323)**								
Salaried employment	455.77 ± 40.84	463.37 ± 41.26	7.59 ^#^	1338 (58.02)	968 (41.98)	2306		0.26
Other works	433.38 ± 39.23	459 ± 43.79	25.61 ^#^	10 (76.92)	3 (23.08)	13		
**Marital status (*n* = 2318)**								
Never married	450.47 ± 42.48	461.06 ± 43.20	10.59 ^#^	271 (62.59)	162 (37.41)	433	6.46	0.04 *
Not living with spouse	454.53 ± 41.69	467.82 ± 44.22	13.29 ^#^	33 (67.35)	16 (32.65)	49		
Living with spouse	457.02 ± 40.24	463.94 ± 40.58	6.92 ^#^	1044 (56.86)	792 (43.14)	1836		

LSTCP—Life Skills Training and Counselling Service Program, SD—Standard Deviation, χ2—test statistics of chi-square test for categorical variables, ^$^—paired t-test, *—*p* value for chi-square test for categorical variables/Fisher’s exact test for categorical variables and paired *t*-test for continuous variable significant at 5%, ^#^ indicates that *p* value for mean difference is significant at 5%.

**Table 3 behavsci-12-00191-t003:** Pre- and post-score assessment of life skill domains among participants of the LSTCSP (*n* = 2669).

Life Skills Domains	Before the Training (Pre-Test)	After the Training (Post-Test)	*t*-Value	*p*-Value
	Mean (SD)			
Decision making	36.65 (3.97)	36.83 (4.04)	2.50	0.0124 *
Problem solving	53.68 (6.13)	54.70 (6.08)	9.56	<0.001 *
Empathy	48.14 (5.56)	45.02 (5.29)	−30.11	<0.001 *
Self-awareness	41.26 (4.74)	40.44 (4.35)	−9.08	<0.001 *
Communication skills	38.77 (4.35)	39.85 (4.50)	13.11	<0.001 *
Interpersonal relationship skills	72.98 (7.22)	73.88 (7.41)	6.93	<0.001 *
Coping with Emotions	35.98 (3.99)	36.40 (3.97)	5.35	<0.001 *
Coping with stress	34.60 (4.18)	35.61 (4.04)	12.65	<0.001 *
Creative Thinking	53.96 (7.28)	56.21 (6.95)	17.75	<0.001 *
Critical thinking	39.37 (5.08)	39.70 (4.76)	3.66	0.0003 *
Life skills total score	455.30 (40.97)	463.10 (41.39)	10.84	<0.001 *

** p* value < 0.05, *t*—paired *t*-test statistics, SD—Standard Deviation.

**Table 4 behavsci-12-00191-t004:** Change in life skill levels (categories) pre- and post-training (*n* = 2323).

X	Levels	Post-Test				
		Moderate	High	Total	χ2	*p* Value *
		*n* (%)				
**Pre-test**	**Moderate**	437 (60.95)	280 (39.05)	717		<0.001
	**High**	185 (11.52)	1421 (88.48)	1606		
	**Total**	622 (26.78)	1701 (73.22)	2323	19.41	

** p* value < 0.05, χ^2^—McNemar–Bowker test statistics value, *n* (%)—number and percentage.

**Table 5 behavsci-12-00191-t005:** Factors associated with increase in life skills among participants (binary logistic regression).

Factor	Crude Odds Ratio	95% Confidence Interval	*p* Value
**Age (in years)**			
36–50	Reference	Reference	Reference
18–35	1.41	1.18–1.70	<0.001 *
51 and above	0.94	0.72–1.25	0.69
**Gender**			
Male	Reference	Reference	Reference
Female	1.32	1.10–1.58	0.003 *
**Marital status**			
Unmarried	Reference	Reference	Reference
Not living with spouse	1.23	0.66–2.31	0.51
Married	0.79	0.64–0.98	0.03 *
**Education**			
Post-graduation & above	Reference	Reference	Reference
Till Degree/Diploma	1.56	1.15–2.12	0.004 *
**Ever used tobacco**			
Yes	Reference	Reference	Reference
No	1.24	0.95–1.60	0.11
**Ever used alcohol**			
Yes	Reference	Reference	Reference
No	1.18	0.96–1.45	0.107
**Depression screened positive**			
Yes	Reference	Reference	Reference
No	0.53	0.23–1.19	0.125
**Big-5 inventory**			
Extroversion score	1.13	1.03–1.24	0.007 *
Agreeableness score	1.1	1.003–1.198	0.042 *
Conscientiousness score	1.09	1.002–1.18	0.045 *
Neuroticism score	0.87	0.79–0.96	0.004 *
Openness score	0.89	0.73–1.10	0.28
**Quality of life**			
Physical quality of life	1.03	1.02–1.03	<0.001 *
Psychological quality of life	1.03	1.02–1.04	<0.001 *
Social quality of life	1.02	1.01–1.02	<0.001 *
Environmental quality of life	1.02	1.01–1.03	<0.001 *

** p* value significant at <0.05.

**Table 6 behavsci-12-00191-t006:** Factors associated with increase in life skills among participants (multiple logistic regression).

Factor	Crude Odds Ratio	95% Confidence Interval	*p* Value	Adjusted Odds Ratio	95% Confidence Interval	*p* Value
**Age (in years)**						
36–50	Reference	Reference	Reference	Reference	Reference	Reference
18–35	1.41	1.18–1.70	<0.001 *	1.34	1.11–1.62	0.003 *
51 and above	0.94	0.72–1.25	0.69	0.94	0.71–1.25	0.67
**Gender**						
Male	Reference	Reference	Reference	Reference	Reference	Reference
Female	1.32	1.10–1.58	0.003 *	1.39	1.15–1.68	0.001 *
**Education**						
Till Degree/Diploma	Reference	Reference	Reference	Reference	Reference	Reference
Post-graduation and above	1.56	1.15–2.12	0.004 *	1.44	1.05–1.97	<0.001 *
**Quality of life**						
Physical quality of life	1.03	1.02–1.03	<0.001 *	1.02	1.01–1.03	<0.001 *
Psychological quality of life	1.03	1.02–1.04	<0.001 *	1.02	1.01–1.03	<0.001 *

** p* value significant at <0.05.

## Data Availability

Not applicable.

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
