# Peer review of "Effectiveness and Factors Associated with Improved Life Skill Levels of Participants of a Large-Scale Youth-Focused Life Skills Training and Counselling Services Program (LSTCP): Evidence from India"

_behavsci, 2022, doi:10.3390/bs12060191_

Round 1
Reviewer 1 Report
Thanky you for the opportunity to read and review this interesting paper.
In order to check the robustness of their findings, the authors should consider treating dependent (criterion) variable as a continuious one, since it is not completely clear why the binary approach was chosen. Namely, even small increases in life skills scores are considered as a "success", while only a small decreases are considered as a "failure", which is not easy to justify.
The authors stated that "all independent variables significant at 10% level in univariate analysis were included into the final multivariate model". What is a justification of this procedure, why not enter all the predictors in the multivariate model?
There are many typos in the paper, two of them even in the paper title, and some additional ones in the affiliation of authors.
Author Response
Reviewer 1
Behavioural Sciences Journal
Dear Sir/Madam,
Thank you very much for reviewing our paper titled “Effectiveness and factors associated with improved life skill levels of participants of a large scale youth focused Life skills Training and Counselling Services program (LSTCP): Evidence from India”. Your comments and suggestions have enriched our manuscript. We have accepted and responded to your comments in the following page point by point. Reviewers’ comments are in black and our responses are in red. Kindly go through.
Thank you
Sincerely,
Authors
Reviewer 2 Report
- Please specify the inequality and socio geographical context related to these problematic health issues highlighted by the literature review for young people in India (rows 28-33): undernutrition, micronutrient deficiency, overweight, common mental disorders, stress, suicides, tobacco use, harmful alcohol use, hypertension, road traffic injuries, and violence. Are they equally distributed in the population? In which classes/areas are more concentrated? And why? How is thesocio- health situation in the State of Karnataka and specifically of youth?
- Please specify more in depth the goals and the expected results of the Yuvaspandana programme related to the above-mentioned problematic health issues highlighted by the literature. The connection is not enough clear. For example, how self-awareness, empathy, effective communication, interpersonal relationships, decision making, problem solving, and creative/critical thinking are connected with (can positively invert the trends of) undernutrition, micronutrient deficiency, overweight, common mental disorders, stress, suicides, tobacco use, harmful alcohol use, hypertension, road traffic injuries, and violence?
- Please explain better how the quasi-experimental design was conducted? Eg. was a control group formed? If yes, how? And how were people in the programme recruited?
- Please focus the discussion more on young people and the effects of the programme on this specific target in comparison with other age targets.
- Please enrich the conclusions more, so that they are not trivial. Please, also add a reflection on the limitations of the study and indicate possible implementations for the future, both on a research/methodological level and in the contents of the programme.
- Typos in the title: Word “particpants” (participants) and space in the title
- Typos: often there is more space among one word and another one (e.g. row 36, 66, 187). Row 66 there are two parentheses. Please check
Author Response
Reviewer 2
Behavioural Sciences Journal
Dear Sir/Madam,
Thank you very much for reviewing our paper titled “Effectiveness and factors associated with improved life skill levels of participants of a large scale youth focused Life skills Training and Counselling Services program (LSTCP): Evidence from India”. Your comments and suggestions have enriched our manuscript. We have accepted and responded to your comments in the following page point by point. Reviewers’ comments are in black and our responses are in red. Kindly go through.
Thank you
Sincerely,
Authors
Reviewer 3 Report
Thank you very much for the opportunity to review the manuscript entitled „Effectiveness and factors associated with improved life skill levels of particpants of a large scale youth focussed Life skills Training and Counselling Services program (LSTCP): Evidence from India “. The article is of interest for the readership of Behavioral Sciences journal as it discusses the effectiveness and factors associated with a training program for improving life skills in India.
I would like to encourage authors to consider several issues to be improved.
First, please revise the manuscript for minor spelling issues. For example, ”particpants” term from the title should be corrected.
Second, the authors should include information regarding the selection procedure of the participants as it is highly important for studies based on experimental design. Also, the authors need to include some limitations of the study given its design and discuss them in relation to the possibility of cofounders’ existence.
In addition, some information related to the loss of training/study participants over time (e.g., those with the worst outcomes leave the treatment) should be also added as this is also a key issue in experimental approaches.
Third, authors should explain more clearly why they choose to employ the used methods and how they complement each other.
I hope that my comments are useful for authors, as they further develop the manuscript.
Author Response
Reviewer 3
Behavioral Sciences Journal
Dear Sir/Madam,
Thank you very much for reviewing our paper titled “Effectiveness and factors associated with improved life skill levels of participants of a large scale youth focused Life skills Training and Counselling Services program (LSTCP): Evidence from India”. Your comments and suggestions have enriched our manuscript. We have accepted and responded to your comments in the following page point by point. Reviewers’ comments are in black and our responses are in red. Kindly go through.
Thank you
Sincerely,
Authors
Round 2
Reviewer 3 Report
Taking into account the revisions and the answer provided by the authors to my comments, I believe that the manuscript has been improved and now warrants publication.

Author Response
Thank you very much. Definitely all the comments and suggestions have improved the manuscript.
